# Stevioside Activates AMPK to Suppress Inflammation in Macrophages and Protects Mice from LPS-Induced Lethal Shock

**DOI:** 10.3390/molecules26040858

**Published:** 2021-02-06

**Authors:** Fuyao Wei, Hong Zhu, Na Li, Chunlei Yu, Zhenbo Song, Shuyue Wang, Ying Sun, Lihua Zheng, Guannan Wang, Yanxin Huang, Yongli Bao, Luguo Sun

**Affiliations:** 1National Engineering Laboratory for Druggable Gene and Protein Screening, Northeast Normal University, Changchun 130024, China; weify753@nenu.edu.cn (F.W.); zhuh897@nenu.edu.cn (H.Z.); lin959@nenu.edu.cn (N.L.); yucl885@nenu.edu.cn (C.Y.); songzb484@nenu.edu.cn (Z.S.); wangsy171@nenu.edu.cn (S.W.); suny040@nenu.edu.cn (Y.S.); huangyx356@nenu.edu.cn (Y.H.); baoyl800@nenu.edu.cn (Y.B.); 2Institute of Genetics and Cytology, Northeast Normal University, Changchun 130024, China; zhenglh015@nenu.edu.cn (L.Z.); wanggn258@nenu.edu.cn (G.W.)

**Keywords:** stevioside, anti-inflammation, AMPK, IRF5, NF-κB, lethal shock

## Abstract

Stevioside, a diterpenoid glycoside, is widely used as a natural sweetener; meanwhile, it has been proven to possess various pharmacological properties as well. However, until now there were no comprehensive evaluations focused on the anti-inflammatory activity of stevioside. Thus, the anti-inflammatory activities of stevioside, both in macrophages (RAW 264.7 cells, THP-1 cells, and mouse peritoneal macrophages) and in mice, were extensively investigated for the potential application of stevioside as a novel anti-inflammatory agent. The results showed that stevioside was capable of down-regulating lipopolysaccharide (LPS)-induced expression and production of pro-inflammatory cytokines and mediators in macrophages from different sources, such as IL-6, TNF-α, IL-1β, iNOS/NO, COX2, and HMGB1, whereas it up-regulated the anti-inflammatory cytokines IL-10 and TGF-β1. Further investigation showed that stevioside could activate the AMPK -mediated inhibition of IRF5 and NF-κB pathways. Similarly, in mice with LPS-induced lethal shock, stevioside inhibited release of pro-inflammatory factors, enhanced production of IL-10, and increased the survival rate of mice. More importantly, stevioside was also shown to activate AMPK in the periphery blood mononuclear cells of mice. Together, these results indicated that stevioside could significantly attenuate LPS-induced inflammatory responses both in vitro and in vivo through regulating several signaling pathways. These findings further strengthened the evidence that stevioside may be developed into a therapeutic agent against inflammatory diseases.

## 1. Introduction

Stevioside, a natural diterpenoid glycoside, has been approved for use as a food sweetener additive by The Food and Agricultural Organization and the World Organization Expert Committee since 2006 [1]. Applications of stevioside in the food industry have been commercialized in Brazil, Canada, China, Japan, Paraguay, the European Union, and the United States [2,3]. Apart from research on the advantages of stevioside compared with traditional food sweeteners, increasing research is focused on the potential pharmacological activities of stevioside [4,5,6,7,8]. Many therapeutic benefits of stevioside have been discovered; it can work as a hypoglycemic, anti-hypertensive, anti-fibrosis, and anti-inflammatory [5,6,8,9,10,11,12,13,14]. A comprehensive evaluation including molecular mechanisms on the anti-inflammatory activity of stevioside has not yet been completed.

Inflammation, a complicated physiological phenomenon, is a natural defensive response to injury, infection, and stress. However, aberrant, uncontrolled inflammation will lead to harmful or even fatal consequences. Acute or chronic inflammatory diseases are serious threats to human health and life [15]. Chronic inflammation, in particular, is related to a wide variety of diseases, including autoimmune diseases, type 2 diabetes, cancers, and so on. Growing prevalence of these inflammatory-related diseases means they are now among the leading causes of morbidity worldwide [16]. Therefore, development of drugs to control inflammation is critical for the prevention or treatment of these diseases. Inflammation results in the production of a variety of pro-inflammatory cytokines such as interleukin (IL)-6, tumor necrosis factor (TNF)-α, and IL-1β, as well as other inflammatory mediators, such as nitric oxide synthase (iNOS)/nitric oxide (NO), cyclo-oxygenase-2 (COX2)/prostaglandin E2 (PGE2), and high mobility group box 1 (HMGB1) [17,18]. These factors mediate and amplify the inflammatory response and are involved in the pathogenesis of many related disorders [17]. In addition to pro-inflammatory cytokines, anti-inflammatory cytokines are produced to balance the inflammatory responses. IL-10 and transforming growth factor-β1 (TGF-β1) are two well-studied regulatory cytokines. Numerous studies have shown that IL-10 treatment can inhibit cytokine production and attenuate the severity of inflammation and TGF-β1 restrains the proliferation and activation of innate immune cells [19,20]. The expression of inflammatory cytokines and mediators is highly regulated by the transcription factor nuclear factor kappa-B (NF-κB). Upon exposure to inflammatory stimuli such as lipopolysaccharide (LPS), NF-κB is activated through IκB kinase (IKK)-mediated phosphorylation, ubiquitination, and subsequent degradation of inhibitor protein of NF-κB (IκB) and then translocates into the nucleus where it facilitates the transcription of target genes [21,22]. In addition to NF-κB, interferon-regulatory factor 5 (IRF5) is also involved in regulating the expression of pro-inflammatory cytokines [23]. IRF5 is a specific marker of inflammatory macrophages and plays important roles in defining the classical inflammatory phenotype of macrophages [24]. LPS, by binding to Toll-like receptor 4, can activate IRF5, which then translocates into the nucleus to drive the expression of target genes. Recent studies have shown that the NF-κB pathway has extended crosstalk with IRFs [25]. The AMP-activated protein kinase (AMPK) pathway is a negative regulator of inflammation. AMPK activation can inhibit NF-κB, suppress the expression of inflammatory genes, and attenuate inflammatory injury [26,27]. Thus, these pathways may be targets of pharmaceutics to control inflammation.

As mentioned above, the anti-inflammatory action of stevioside is most likely to be a complex process that must be fully elucidated for potential applications of stevioside as a novel therapeutic agent for inflammation. Therefore, in this study, we comprehensively evaluated the anti-inflammatory capacity of stevioside and systemically investigated its underlying molecular mechanisms in three sources of macrophages and in mice.

## 2. Results

### 2.1. Stevioside Inhibited IL-6 Expression in LPS-Stimulated Macrophages

Stevioside, a natural compound, was found to have significant inhibitory effects on the IL-6 promoter activity in luciferase reporter assays (data not shown). We then confirm its inhibitory effects on LPS-induced IL-6 expression in macrophages from different sources, mouse macrophage cell line RAW 264.7, human monocytic leukemic cell line THP-1, and primary mouse peritoneal macrophages. Before we started, effects of stevioside on macrophage viability were determined to exclude the possibility that stevioside would contribute to cytotoxicity of macrophages. The MTT assay showed that LPS alone or LPS combined with stevioside up to 400 μg/mL did not affect the macrophage viability significantly (Figure 1a). According to previous research [11,12,14], three doses of stevioside, 25 μg/mL, 50 μg/mL, and 100 μg/mL, were used in in vitro experiments. Then, we verified whether stevioside could down-regulate IL-6 expression in LPS-stimulated macrophages from the three different sources. In macrophages of all three sources, results showed that stevioside repressed LPS-induced IL-6 expression at both the mRNA level as assessed by RT and qRT-PCR (Figure 1b,c) and protein level as detected by ELISA and Western blot (Figure 1d,e). Moreover, stevioside showed inhibitory effects on IL-6 expression in a dose-dependent manner when tested in RAW 264.7 cells (Figure 1e). These results indicated that stevioside inhibited the expression of IL-6 transcriptionally, in agreement with results of previous luciferase reporter assays (data now shown).

### 2.2. Stevioside Repressed LPS-Induced Expression of Pro-Inflammatory Cytokine TNF-α and IL-1β in Macrophages

Next, we investigated the possible effects of stevioside on the expression of other pro-inflammatory cytokines in macrophages exposed to LPS. RT-PCR and RT-qPCR analysis revealed that LPS alone significantly enhanced the mRNA expression of TNF-α and IL-1β, which were suppressed by pretreatment with stevioside in macrophages (Figure 2a,b,d). Then, we further confirmed that pretreatment with stevioside also inhibited LPS-induced production of TNF-α and IL-1β protein in macrophages by ELISA (Figure 2c,e). However, the inhibition on IL-1β expression did not reach statistical significance in some types of macrophages. These results indicated that stevioside suppressed LPS-induced pro-inflammatory cytokine expression.

### 2.3. Stevioside Enhanced Expression of Anti-Inflammatory Cytokines in LPS-Stimulated Macrophages

Since stevioside suppressed LPS-induced expression of representative pro-inflammatory cytokines, we wondered whether stevioside had any effects on anti-inflammatory cytokine expression. First, we examined IL-10 expression in LPS-induced macrophages pretreated with stevioside. The results showed that pretreatment with stevioside could enhance the LPS-induced mRNA expression (Figure 3a) and production of IL-10 (Figure 3b) in macrophages from the three sources. Similarly, we also found that stevioside pretreatment could augment the mRNA expression of TGF-β1 in the three sources of macrophages (Figure 3c) and protein products in dTHP-1 cells and murine peritoneal macrophages; the production of TGF-β1 was increased but did not reach statistical significance in RAW 264.7 cells (Figure 3d) compared to that with LPS treatment alone. Taken together, these results implied that stevioside was also involved in regulating the expression of anti-inflammatory cytokines.

### 2.4. Stevioside Suppressed LPS-Induced iNOS, COX-2, and HMGB1 Expression in Macrophages

Many pro-inflammatory mediators contribute to the pathogenesis of inflammation, such as iNOS/NO, COX2/PGE2, and HMGB1, and their elevated levels are also characteristic of inflammation. Therefore, we further examined the effects of stevioside on these inflammatory mediators. We detected the expression level of iNOS, the enzyme responsible for producing NO. The results showed that LPS alone significantly enhanced iNOS mRNA levels in three sources of macrophages (Figure 4a,b) using both RT-PCR and RT-qPCR analysis. However the enhancement effects were nearly completely abolished by stevioside pretreatment. Consistently, the NO levels in the supernatant of these three types of macrophages were increased upon LPS induction, which was also suppressed in the presence of stevioside at 100 μg/mL (Figure 4c). In addition, stevioside was found to reduce the protein level of COX-2 in a dose-dependent manner in mouse peritoneal macrophages (Figure 4d). As for HMGB1, the results clearly showed that stevioside decreased extracellular HMGB1 levels of LPS-treated macrophages from the three sources (Figure 4e). Thus, stevioside showed similar inhibitory effects on LPS-induced production of the detected pro-inflammatory mediators.

### 2.5. Stevioside Repressed LPS-Induced NF-κB and IRF5 Activation in Macrophages

Stevioside not only suppressed the LPS-induced expression of various pro-inflammatory mediators and cytokines, but also promoted the expression of anti-inflammatory cytokines IL-10 and TGF-β1. These findings suggested that stevioside had anti-inflammatory activities at least against LPS-mediated inflammation. Thus, we wanted to investigate the mechanisms by which stevioside exerts its effects.

NF-κB plays a critical role in regulating the expression of pro-inflammatory factors. Our results showed that pretreatment with stevioside attenuated the LPS-induced nuclear translocation of p65 in RAW 264.7 cells (Figure 5a) and inhibited phosphorylation of p65 in mouse peritoneal macrophages (Figure 5b) in a dose-dependent manner. Next, we explored whether stevioside could inhibit the LPS-induced phosphorylation and degradation of IκB-α in RAW 264.7 macrophages. The results showed that the LPS-induced phosphorylation and degradation of IκB-α were significantly blocked by stevioside pretreatment (Figure 5c).

In addition, we examined the effects of stevioside on the LPS-induced activation of IKK-α in RAW 264.7 macrophages and mouse peritoneal macrophages. Stevioside was found to markedly reduce LPS-induced IKK-α phosphorylation but did not affect the total amounts of IKK-α (Figure 6). These data confirmed that stevioside exerted its inhibitory effects by repressing LPS-triggered activation of the NF-κB pathway.

Next, we wondered whether stevioside would also influence IRF5 activation. We found that stevioside not only decreased total IRF5 expression levels (Figure 7a) in both RAW 264.7 cells and mouse peritoneal macrophages, but it also notably decreased the level of nuclear IRF5 induced by LPS in a dose-dependent manner (Figure 7b). This result suggested that stevioside suppressed both IRF5 expression and its activation.

### 2.6. Stevioside Inhibited IRF5/NF-κB Pathway through AMPK Activation

To further elucidate the mechanisms underlying the activities of stevioside, its effects on the AMPK pathway, a negative regulator of inflammation, was investigated. Stevioside was found to enhance phosphorylation of AMPK in LPS-treated RAW 264.7 cells and mouse peritoneal macrophages at 100 μg/mL (Figure 8a,b). Moreover, we examined the correlation between stevioside-induced AMPK activation and its inhibition of the IRF5 and NF-κB pathways. RAW 264.7 cells were pretreated with the AMPK inhibitor compound C (CC) at concentrations of 3.125, 6.25, and 12.5 μM for 1 h. We found that 12.5 μM CC could effectively abolish the induced activation of AMPK as shown by the downregulation of phosphorylated AMPK (Figure 8c). Treatment with 12.5 μM CC also increased nuclear translocation of P65 and IRF5 in RAW 264.7 cells exposed to LPS plus stevioside (Figure 8d). These results suggested that inhibition of AMPK activation attenuated the inhibitory effects of stevioside on IRF5 and NF-κB activation.

### 2.7. Stevioside Attenuated the Inflammatory Response and Protected Mice from Excessive Cytokine-Mediated Lethal Shock

Based on the in vitro studies, we then evaluated the in vivo effects of stevioside in a mouse model of LPS-induced excessive cytokine-mediated lethal shock. According to the previous study and the conversion coefficient calculation method, we selected 10, 20, and 40 mg/kg stevioside to be used in our in vivo experiments [10,28]. In our system, mice did not develop lethal shock when given D-GalN alone even up to 15 mg per mouse, while i.p. injection of 5 μg/kg LPS in D-GalN-sensitized mice led to 100% mortality (*n* = 10) at 12 h post injection. However, when LPS was injected together with stevioside at 10, 20, and 40 mg/kg, the rate of lethality in mice was reduced to 80%, 60%, and 40%, respectively, at 60 h post injection (Figure 9a). Moreover, we tested the suitable time point for administration of stevioside in D-GalN/LPS-treated mice. Stevioside (40 mg/kg) was i.p. injected into D-GalN-sensitized mice simultaneously with, 1 h before, or 1 h after LPS injection. The results showed that stevioside could effectively protect mice from LPS-induced lethal shock when injected at −1 h or simultaneously with LPS injection, but it failed to display the protective role when injected after LPS injection (Figure 9b).

Next, we investigated the effects of stevioside on cytokines and pro-inflammatory mediators in the lethal shock mouse model. LPS administration (5 μg/kg i.p.) markedly increased serum levels of IL-6, TNF-α, IL-1β, HMGB1, and NO, but LPS together with stevioside (40 mg/kg i.p., *n* = 10) administration significantly decreased the production of these three pro-inflammatory cytokines, as well as HMGB1 and NO (Figure 10a). By contrast, stevioside increased the serum level of IL-10 (Figure 7c). These results in vivo are consistent with those observed in vitro. The stevioside-mediated down-regulation of pro-inflammatory factors and up-regulation of anti-inflammatory cytokines may contribute to the survival of mice with lethal shock. In agreement with the in vitro data, we observed that stevioside could obviously activate the AMPK pathway in periphery blood mononuclear cells of some of mice with lethal shock (6 of 10 mice) (Figure 10b).

In addition, pathological analysis revealed that LPS induced severe inflammatory cell infiltration and liver hemorrhage and necrosis in d-GalN-sensitized mice, whereas stevioside could alleviate all these pathological changes of the liver (Figure 11), implying that stevioside carried out an inhibitory role on LPS-induced liver inflammation and injury in mice. Together, these results suggested that stevioside could elicit its anti-inflammatory effects both in vitro and in vivo.

## 3. Discussion

In the present study, stevioside was validated to be capable of inhibiting LPS-induced expression of pro-inflammatory factors such as IL-6, TNF-α, IL-1β, NO, COX2, and HMGB1 and promoting the production of anti-inflammatory cytokines in macrophages from three sources, and it also elicited its anti-inflammatory effects in mice and protected mice from excessive cytokine-mediated lethal shock. These anti-inflammatory effects of stevioside might be related to its activation on the AMPK pathway, which in turn inhibited NF-κB/IRF5-mediated inflammation.

Stevioside has been demonstrated to suppress LPS-induced inflammatory cytokine secretion in RAW 264.7 cells in previous reports [11,12]. In order to comprehensively investigate its anti-inflammatory effects, we carried out studies in macrophages of three sources in vitro combined with in vivo experiments. The source of the macrophages might affect their behavior and the interpretation of results. Immortalized cell lines may exhibit functional defects compared to primary cells due to abnormal genetic structures, while multiple functional differences exist between macrophages from different species [29]. Here, we used three sources of macrophages, mouse- or human-derived macrophage cell lines, RAW 264.7 and THP-1, and primary mouse peritoneal macrophages. Most of the anti-inflammatory effects of stevioside are commonly observed in all three sources of macrophages, for example, inhibiting LPS-induced expression of pro-inflammatory cytokines IL-6 and TNF-α (Figure 1 and Figure 2) and other inflammation-associated mediators, such as iNOS, NO, COX-2, and HMGB1 (Figure 3). However, suppression of LPS-induced IL-1β by stevioside with statistical significance was only be observed in dTHP-1 cells at both mRNA and protein levels (Figure 3d,e). However, in LPS-treated mice, IL-1β induction and the induction of other pro-inflammatory factors were effectively inhibited by stevioside (Figure 10a). Therefore, combination of in vitro and in vivo data further confirmed that stevioside could extensively suppress LPS-induced inflammatory responses and might serve as a promising anti-inflammatory agent.

While IL-6, TNF-α, and IL-1β are involved in mediating inflammatory responses, IL-10 and TGF-β1 are well known anti-inflammatory cytokines that inhibit the secretion of pro-inflammatory cytokines and attenuate inflammation [30,31]. In our study, stevioside was demonstrated to be capable of enhancing IL-10 and TGF-β1 expression to some extent in LPS-treated macrophages from the three sources and in LPS-treated mice as well. Therefore, we speculated that induction of anti-inflammatory factors might contribute to the anti-inflammatory effects of stevioside. Moreover, we confirmed that stevioside inhibited LPS–induced activation of the NF-κB pathway in macrophages, consistent with previous findings in RAW 264.7 and THP-1 cells [11,12,14]. In addition, we demonstrated that stevioside also inhibited the expression of IRF5, another transcription factor that promotes the transcription of pro-inflammatory cytokines, and down-regulated its level in the nucleus. Meanwhile, our study showed that stevioside obviously enhanced activation of AMPK, a negative regulator of pro-inflammation. Pre-treatment with the AMPK inhibitor CC completely abolished the suppressive effect of stevioside on the LPS-induced activation of NF-κB and IRF5. That finding suggested that enhancement of AMPK activation may mediate the inhibitory effects of stevioside on NF-κB/IRF5 pathways. Taken together, the results indicate that stevioside exerts anti-inflammatory effects through versatile mechanisms.

In agreement with its anti-inflammatory effects observed in vitro, stevioside also exhibited activities against LPS-induced inflammation in vivo. LPS induces rapid release of pro-inflammatory cytokines and mediators into the blood and leads to excessive cytokine-mediated liver injury and lethal shock in d-GalN-sensitized mice. By using this murine model, we demonstrated that treatment with stevioside could dramatically reduce the serum levels of IL-6, TNF-α, IL-1β, HMGB1, and NO and increase the survival rates of animals with lethal shock. Consistently, stevioside reduced the infiltration of inflammatory cells and damage in the liver. The maximum dose of stevioside used in the murine model (40 mg/kg) was converted into an equivalent dose for humans as 263.7 mg/kg by using the conversion coefficient calculation method; this value is much lower than the no-observed-adverse-effect dose (970 mg/kg) provided by The Joint FAO/WHO Expert Committee [7,28]. Therefore, we speculate that the in vivo concentration of stevioside used in our study suggests an anti-inflammatory effect in humans. Moreover, stevioside activated AMPK in periphery blood mononuclear cells from the model mice, indicating that activation of AMPK might be related to the anti-inflammatory effects of stevioside in vivo. The protective effects of stevioside on LPS-induced lethal shock mice only occurred when injected before or simultaneously with LPS but not after LPS injection. This may hint that stevioside competed with LPS on target cells, which remains to be further explored, as well as its specific targets.

## 4. Materials and Methods

### 4.1. Materials

Polyclonal antibodies against IL-6, COX-2, p65 (NF-κB), histone 1, IκB-α, IKK, and IRF5 were obtained from Santa Cruz Biotechnology (Santa Cruz, CA, USA). A mouse monoclonal antibody against glyceraldehyde 3-phosphate dehydrogenase (GAPDH) was purchased from Kang Cheng Biotech (Shanghai, China). Anti-phospho-p65, anti-phospho-IκB, anti-phospho-IKK, anti-AMPK, and anti-phospho-AMPK were all obtained from Cell Signaling Technology (Beverly, MA, USA). LPS, phorbolmyristate acetate (PMA), compound C (CC, 6-1-3-pyridin-4-ylpyrazolo-1-pyrimidine), and d-glactosamine (d-GalN) were obtained from Sigma-Aldrich (St. Louis, MO, USA). Enhanced chemiluminescence (ECL) reagent was purchased from Beyotime (Shanghai, China). Mouse/human enzyme-linked immunosorbent assay (ELISA) kits for IL-6, TNF-α, IL-1β, HMGB1, IL-10, and TGF-β1 were obtained from R&D Systems (Beijing, China). The NO kit was purchased from Beyotime. Stevioside (HPLC ≥ 98%) was purchased from the National Institute for the Control of Pharmaceutical and Biological Products (NICPBP, Beijing, China) and was dissolved in PBS.

### 4.2. Cell Culture

RAW 264.7 cells were obtained from the Chinese Academy of Sciences Shanghai Institute for Biological Sciences Cell Resource Center. Cells were maintained in Dulbecco’s modified Eagle’s medium (DMEM, GIBCO BRL, Grand Island, NY, USA.) supplemented with 10% fetal bovine serum (FBS, TBD, Beijing, China) and antibiotics (100 U/mL penicillin and 100 μg/mL streptomycin, Ameresco, Framingham, MA, USA). THP-1 cells were cultured in RPMI 1640 medium (Hyclone, Logan, UT, USA) supplemented with 10% FBS and antibiotics and were induced to differentiate (dTHP-1) upon exposure to 0.2 μg/mL PMA for 24 h. Mouse peritoneal macrophages were obtained from ascites of C57BL/6J mice that were injected intraperitoneally (i.p.) with 1 mL of 5% soluble starch 3 days before sacrifice. The cells were washed twice and re-suspended in DMEM with 5% FBS. All cells were cultured in a 37 °C incubator with 5% CO_2_.

### 4.3. Animals

Female BALB/C mice and C57BL/6 mice (eight weeks old, about 20 g) were purchased from the Experimental Animal Center, Medical College of Norman Bethune, Jilin University and were housed under pathogen-free conditions.

### 4.4. RNA Extraction, RT-PCR, and Real-Time RT-PCR (RT-qPCR)

Total RNA was prepared from cultured cells using Trizol reagent (Invitrogen, Carlsbad, CA, USA) following the manufacturer’s instructions. The RT-PCR kit was purchased from TransGen Biotech (TransGen Biotech, Beijing, China). Total RNA (3 μg) was reverse transcribed into cDNA according to instructions. The target genes were then amplified by RT-PCR or RT-qPCR with corresponding primers (Table 1) as described previously [32].

### 4.5. ELISA

IL-6, TNF-α, IL-1β, HMGB1, IL-10, and TGF-β1 levels in cell culture media and mouse sera were quantified by using ELISA kits according to the manufacturer’s instructions (R&D Systems, Beijing, China).

### 4.6. NO Determination

NO levels in cell culture media and mouse sera were quantified by using the NO assay kit according to the manufacturer’s instructions (Beyotime Biotechnology, Shanghai, China).

### 4.7. Protein Extraction and Western Blotting Analysis

Whole cell extracts or cytosolic and nuclear extracts were prepared as previously described [32,33]. The extracts were then resolved by 12% sodium dodecyl sulfate polyacrylamide gel electrophoresis (SDS-PAGE), transferred to a polyvinylidene fluoride (PVDF) membrane, and sequentially probed with various primary antibodies and horseradish peroxidase (HRP)-conjugated secondary antibodies for 1 h at room temperature. Protein bands were visualized using ECL. Western blotting data were quantified using Image J software.

### 4.8. Lethal Shock Model

The mouse model of excessive cytokine-mediated lethal shock was constructed as reported previously [34]. Briefly, BALB/C mice were first injected i.p. with PBS (as control) or 750 mg/kg d-GalN (in PBS), followed by i.p. injection of LPS (5 μg/kg) after 1.5 h. Mice were then randomly divided into different groups and further i.p. injected with stevioside or saline (as control) at different concentrations and at different time points as indicated. Mouse survival was monitored for 60 h. To obtain sera or peripheral blood cells, mice were anaesthetized with ether 4 h after the LPS injection and then blood was collected with or without heparin by quickly removing eyeballs. Sera from agglutinative blood was collected for ELISA or NO assay. Periphery blood mononuclear cells were harvested from nonagglutinative blood by removing red blood cells. Then, mononuclear cells were lysed to obtain protein samples for Western blotting. For histological analysis, all mice were killed under ether anesthesia when the first death occurred.

### 4.9. Histological Analysis of Mouse Livers

Livers of mice were fixed in 10% neutral buffered formalin, embedded in paraffin, cut into 5–10 μm thickness, and then affixed to slides. The sections were stained with hematoxylin and eosin (H&E). Morphological changes in the sections of liver tissue were observed under a microscope.

### 4.10. Statistical Analysis

Experiments were repeated at least three times with two replicates per sample for each experiment. The student *t*-test was used to calculate the statistical significance of the experimental results. The significance was set as * *p* < 0.05, ** *p* < 0.01, *** *p* < 0.001 versus LPS-treated group and # *p* < 0.05, ## *p* < 0.01, ### *p* < 0.001 versus control group. Error bars denote the standard deviation (S.D.).

## 5. Conclusions

In summary, stevioside may function as an inhibitor of inflammation through multiple mechanisms. The activation of the AMPK pathway, which in turn inhibits NF-κB/IRF5-mediated inflammation might be involved. As a natural food additive, stevioside is a perfect candidate for anti-inflammatory medicine development in terms of safety. Accordingly, stevioside might be a potential treatment option for inflammatory diseases.

## Figures and Tables

**Figure 1 molecules-26-00858-f001:**
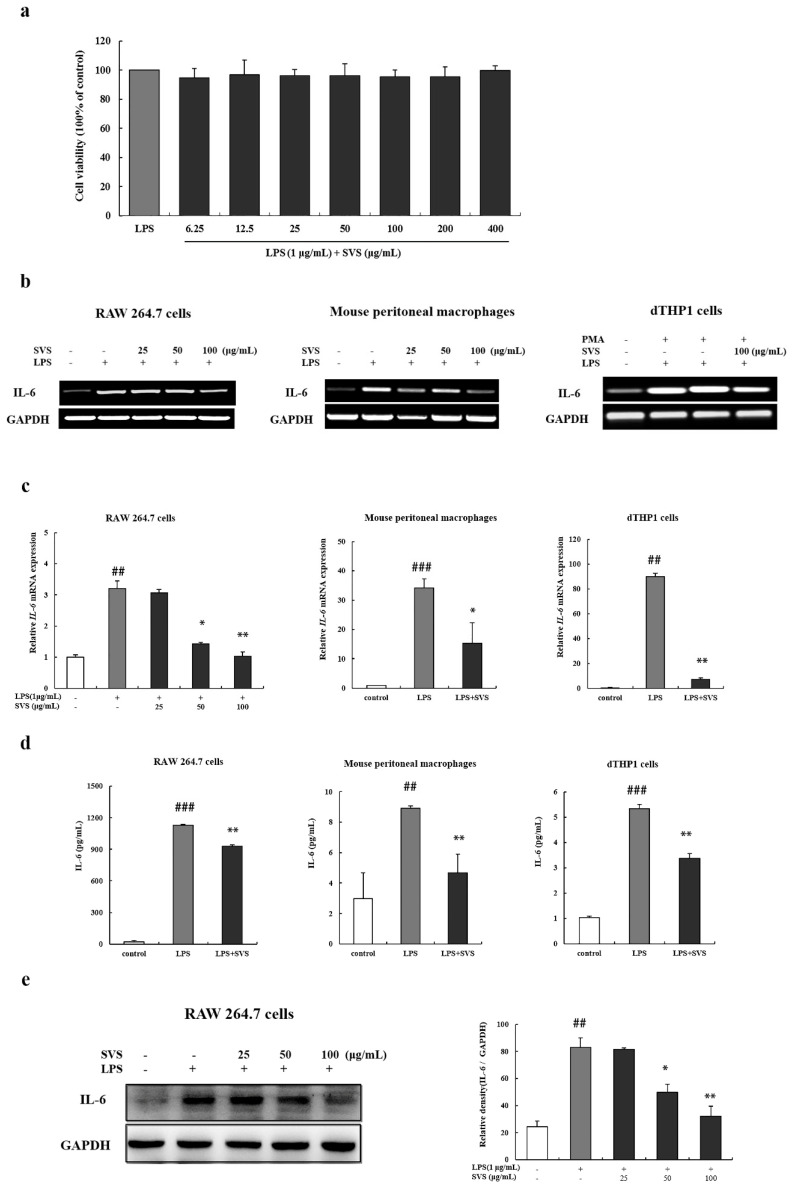
Stevioside inhibited IL-6 expression in lipopolysaccharide (LPS)-induced macrophages. (**a**) Effect of stevioside on the viability of RAW 264.7 cells. Cells were treated with or without stevioside at 0, 6.25, 12.5, 25, 50, 100, 200, and 400 μg/mL with LPS (1 μg/mL). At 24 h post-treatment, 20 μL MTT was added to each well for an additional 4 h of incubation. Cell viability was expressed as a percentage of the control; (**b**,**c**) Macrophages were pretreated with or without stevioside (25, 50, or 100 μg/mL) for 1 h and then stimulated with LPS (1 μg/mL) for 9 h. RT-PCR (**b**) and RT-qPCR analysis (**c**) were used to detect and quantify IL-6 transcripts; (**d**) Macrophages were pretreated with or without stevioside (100 μg/mL) for 1 h followed by LPS (1 μg/mL) stimulation for another 24 h. IL-6 levels in the culture supernatant were determined by ELISA; (**e**) RAW 264.7 cells were pretreated with or without stevioside (25, 50, or 100 μg/mL) for 1 h and then stimulated with LPS (1 μg/mL) for 12 h. IL-6 expression was detected by Western blot. GAPDH was used as an internal control. Quantification of IL-6 levels is shown in the left panel, as measured by image J. Experiments were repeated three times. ## *p* < 0.01, ### *p* < 0.001 versus control group; * *p* < 0.05; ** *p* < 0.01 versus LPS-treated group. SVS, stevioside.

**Figure 2 molecules-26-00858-f002:**
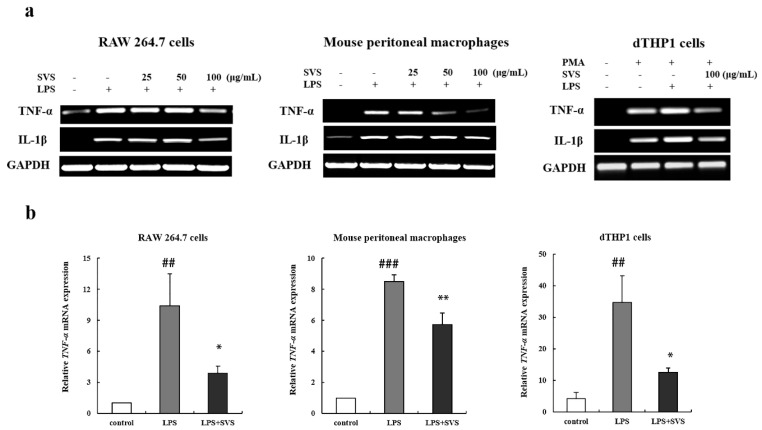
Stevioside reduced production of TNF-α and IL-1β in LPS-stimulated macrophages. (**a**) Transcripts of TNF-α and IL-1β was examined by RT-PCR in macrophages pretreated with or without stevioside at the indicated concentrations followed by LPS (1 μg/mL) stimulation for 9 h. GAPDH was used as an internal control; (**b**,**d**) mRNA levels of TNF-α and IL-1β were quantified by RT-qPCR in macrophages pretreated with or without 100 μg/mL stevioside followed by LPS (1 μg/mL) stimulation for 9 h; (**c**,**e**) Protein levels of TNF-α and IL-1β were detected by ELISA in the supernatant of macrophages with stevioside (100 μg/mL) pretreatment for 1 h and LPS (1 μg/mL) stimulation for another 24 h. Experiments were repeated three times. # *p* < 0.05, ## *p* < 0.01, ### *p* < 0.001 versus control group; * *p* < 0.05; ** *p* < 0.01 versus LPS-treated group; ns, no significant difference versus LPS-treated group. SVS, stevioside.

**Figure 3 molecules-26-00858-f003:**
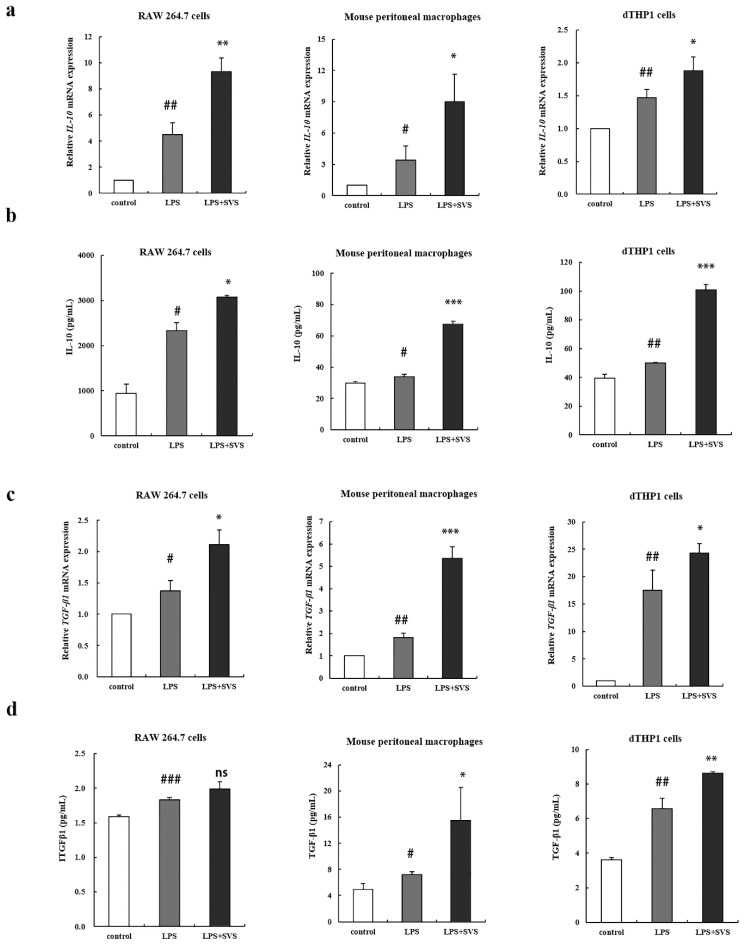
Stevioside enhanced IL-10 and TGF-β1 expression in LPS-stimulated macrophages. Macrophages were pretreated with or without stevioside (100 μg/mL) for 1 h and then stimulated with LPS (1 μg/mL) for another 9 h (**a**,**c**) or 24 h (**b**,**d**); (**a**,**c**) mRNA levels of IL-10 (**a**) and TGF-β1 (**c**) in the cells was detected by RT-qPCR; (**b**,**d**) Protein levels of TNF-α (**b**) and IL-1β (**d**) were detected by ELISA in the supernatant of macrophages. Experiments were repeated three times. # *p* < 0.05, ## *p* < 0.01, ### *p* < 0.001 versus control group; * *p* < 0.05, ** *p* < 0.01, *** *p* < 0.001 versus LPS-treated group. ns, no significant difference versus LPS-treated group. SVS, stevioside.

**Figure 4 molecules-26-00858-f004:**
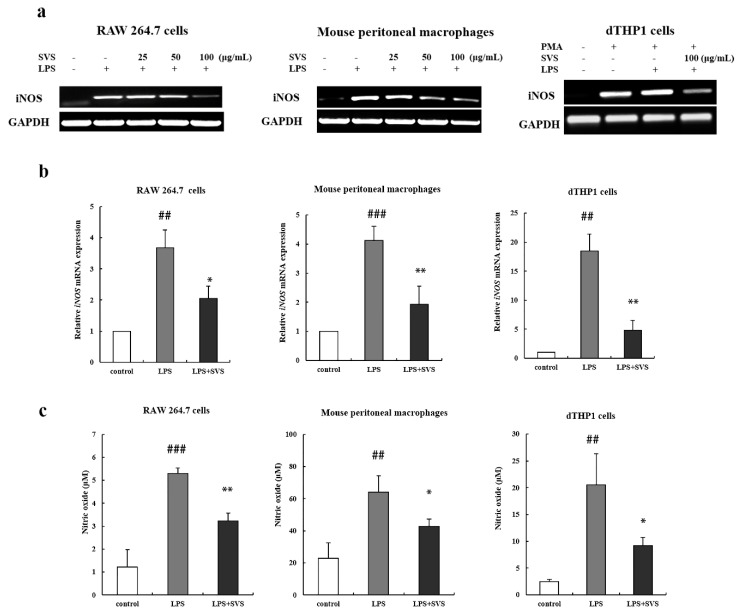
Stevioside suppressed LPS-induced iNOS, COX-2, and HMGB1 expression in macrophages. (**a**,**b**) Macrophages were pretreated with or without stevioside (100 μg/mL) for 1 h and then stimulated with LPS (1 μg/mL) for another 9 h. Transcripts of iNOS were examined by RT-PCR (**a**) or RT-qPCR (**b**) in macrophages; (**c**) Macrophages were pretreated with or without stevioside (100 μg/mL) for 1 h and then stimulated with LPS (1 μg/mL) for another 24 h. The production of NO was detected by the Griess reaction in the culture medium; (**d**) Macrophages were pretreated with or without stevioside (25, 50, or 100 μg/mL) for 1 h and then stimulated with LPS (1 μg/mL) for another 4 h. COX-2 protein was detected by Western blot in cell lysates. GAPDH was used as an internal control; (**e**) Macrophages were treated as in (**c**). HMGB1 level was determined by ELISA in the culture medium; # *p* < 0.05, ## *p* < 0.01, ### *p* < 0.001 versus control group; * *p* < 0.05, ** *p* < 0.01 versus LPS-treated group. SVS, stevioside.

**Figure 5 molecules-26-00858-f005:**
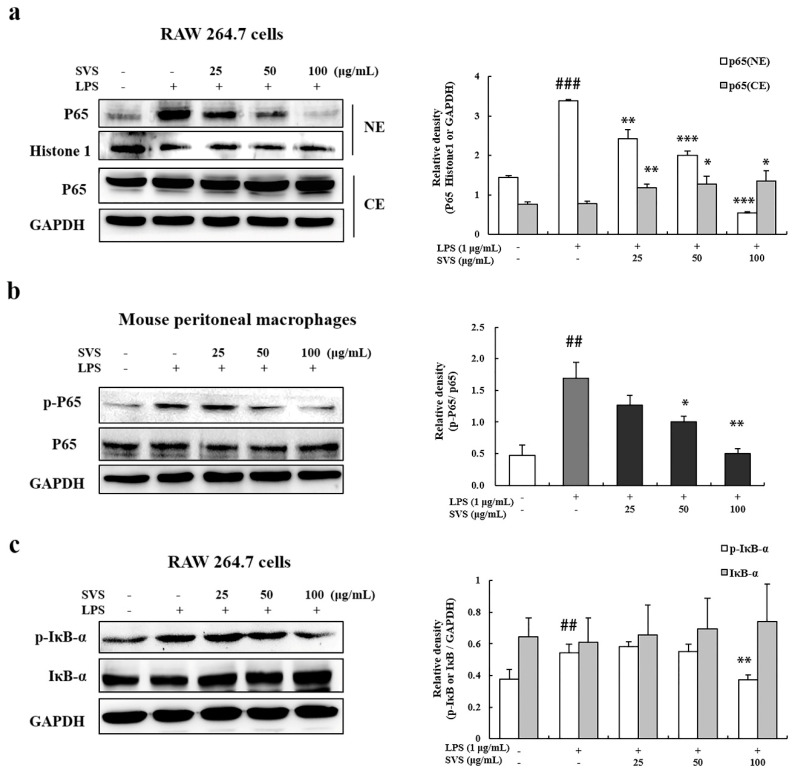
Stevioside repressed LPS-induced NF-κB activation in macrophages. RAW 264.7 cells were pretreated with or without the indicated concentrations of stevioside for 1 h and then stimulated with LPS (1 μg/mL) for 15 min (**c**) or 1 h (**a**,**b**). Nuclear and cytosolic extracts (**a**) or whole cell lysate (**b**,**c**) were isolated and subjected to immunoblots. GAPDH was used as an internal control for whole cell lysate and cytosolic extracts while histone 1 was used as internal control for nuclear extracts. The right panels in a, b, and c are the quantification of their left panels, as measured by Image J. (**a**) p65; (**b**) P65, and phospho-P65 (p-P65); (**c**) IκB-α and phospho-IκB-α (p-IκB-α). Experiments were repeated three times, and similar results were obtained. Data are presented as means ± standard deviation (S.D.) of three independent experiments. ## *p* < 0.01, ### *p* < 0.001 versus control group; * *p* < 0.05, ** *p* < 0.01, *** *p* < 0.001 versus LPS-treated group. SVS, stevioside.

**Figure 6 molecules-26-00858-f006:**
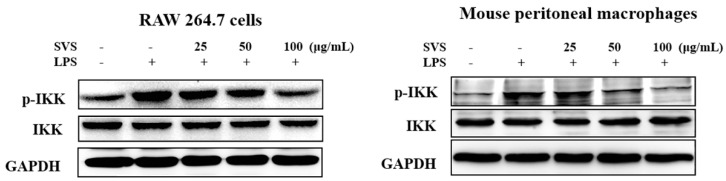
The effects of stevioside on the LPS-induced activation of IKK in RAW 264.7 macrophages and mouse peritoneal macrophages. RAW 264.7 cells were pretreated with or without the indicated concentrations of stevioside for 1 h and then stimulated with LPS (1 μg/mL) for 15 min. Whole cell lysates were isolated and subjected to immunoblots. GAPDH was used as an internal control. SVS, stevioside.

**Figure 7 molecules-26-00858-f007:**
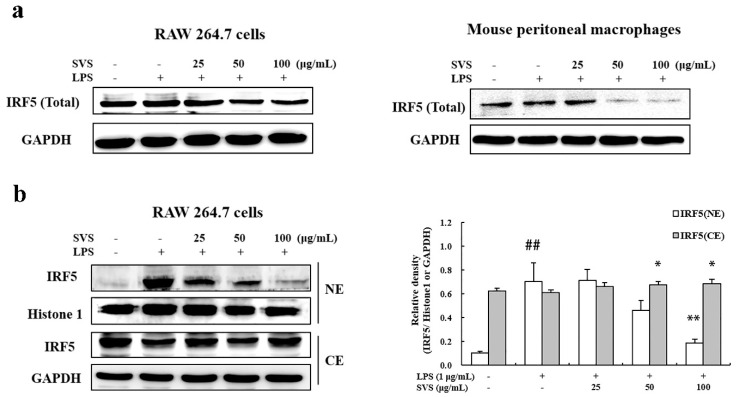
Stevioside repressed LPS-induced IRF5 activation in macrophages. RAW 264.7 cells were pretreated with or without the indicated concentrations of stevioside for 1 h and then stimulated with LPS (1 μg/mL) for 1 h. Whole cell lysate (**a**) or nuclear and cytosolic extracts (**b**) were isolated and subjected to immunoblots. GAPDH was used as an internal control for whole cell lysate and cytosolic extracts while histone 1 was used as internal control for nuclear extracts. The right panel in b is the quantification of the left panel, as measured by Image J. Experiments were repeated three times, and similar results were obtained. Data are presented as means ± S.D. of three independent experiments. ## *p* < 0.01 versus control group; * *p* < 0.05, ** *p* < 0.01 versus LPS-treated group. SVS, stevioside.

**Figure 8 molecules-26-00858-f008:**
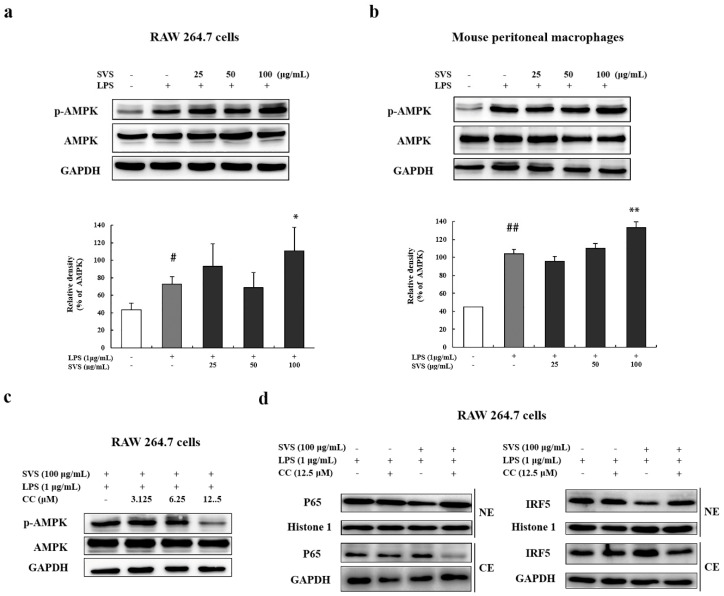
Stevioside inhibited the IRF5/NF-κB pathway through AMPK activation. (**a**,**b**) Protein levels of AMPK and phosphor-AMPK (p-AMPK) were detected by Western blot in RAW 264.7 cells and mouse peritoneal macrophages, which were pretreated with or without stevioside (25, 50, or 100 μg/mL) for 1 h and then stimulated with LPS (1 μg/mL) for 1 h. The lower panels are quantifications of the blots, as measured by Image J; (**c**) Protein levels of AMPK and phosphor-AMPK (p-AMPK) were examined by Western blot in RAW 264.7 cells treated with indicated concentrations of compound C (CC) for 1 h, followed by 100 μg/mL stevioside treatment for 1 h and 1 μg/mL LPS stimulation for another 1 h; (**d**) Protein levels of P65 (left panel) and IRF5 (right panel) were detected by Western blot in nuclear (NE) and cytosolic extracts (CE) of RAW 264.7 cells, which were treated as in (**c**) using 12.5 μM CC. Histone 1 and GAPDH were used as internal controls. # *p* < 0.05, ## *p* < 0.01 versus control group; * *p* < 0.05, ** *p* < 0.01 versus LPS-treated group. SVS, stevioside.

**Figure 9 molecules-26-00858-f009:**
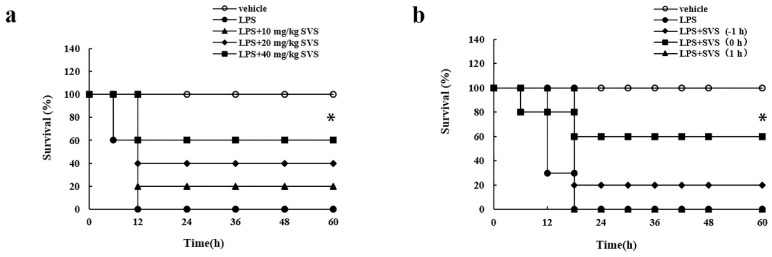
The in vivo effects of stevioside in a mouse model of LPS-induced excessive cytokine-mediated lethal shock. (**a**) Dose-dependent effect of stevioside on survival of mice with lethal shock (*n* = 10); (**b**) Time-dependent effect of stevioside on survival of mice with lethal shock (*n* = 10). * *p* < 0.05 versus LPS-treated group. SVS, stevioside.

**Figure 10 molecules-26-00858-f010:**
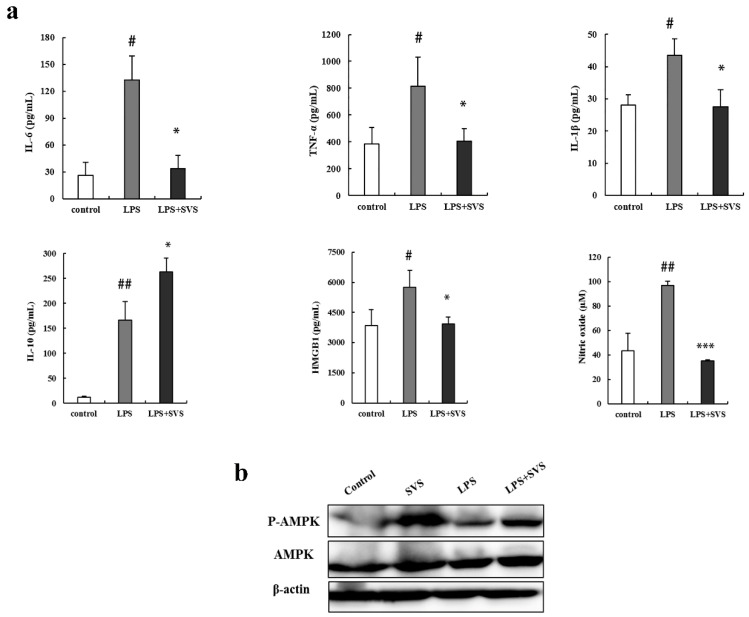
Stevioside attenuated inflammatory responses and protected mice from LPS-induced excessive cytokine-mediated lethal shock. (**a**) Serum levels of IL-6, TNF-α, IL-1β, IL-10, and HMGB1 were detected by ELISA and NO were detected by the Griess reaction; (**b**) Representative immunoblots of AMPK and phosphor-AMPK (p-AMPK) in periphery blood mononuclear cells of mice with lethal shock. # *p* < 0.05, ## *p* < 0.01 versus control group; * *p* < 0.05, *** *p* < 0.001 versus LPS-treated group. SVS, stevioside.

**Figure 11 molecules-26-00858-f011:**
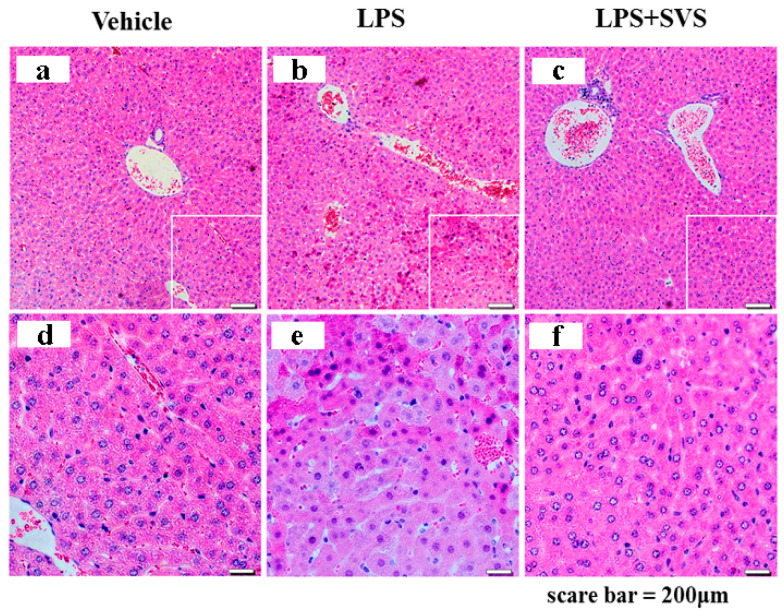
Histological sections of mouse livers. Scale bar represents 200 μm. Panels **d**, **e**, and **f** are images amplified from panels **a**, **b**, and **c**, respectively.

**Table 1 molecules-26-00858-t001:** Mouse and human primer sequences for genes used for RT-PCR and RT-qPCR.

Gene	Forward Primer	Reverse Primer
IL-6 Mouse	5′-TCCAGTTGCCTTCTTGGGAC-3′	5′-GTGTAATTAAGCCTCCGACTTG-3′
Human	5′-GTGAAAGCAGCAAAGAGGCACT-3′	5′-ACCAGGCAAGTCTCCTCATTGA-3′
TNF-α Mouse	5′-GACCCTCACACTCAGATCATCTTCT-3′	5′-CCTCCACTTGGTGGTTTGCT-3′
Human	5′-TCAATCGGCCCGACTATCTC-3′	5′-CAGGGCAATGATCCCAAAGT-3′
IL-1β Mouse	5′-TCGTGCTGTCGGACCCATAT-3′	5′-GTCGTTGCTTGGTTCTCCTTGT-3′
Human	5′-GCACGATGCACCTGTACGAT-3′	5′-AGACATCACCAAGCTTTTTTGCT-3′
TGF-β1 Mouse	5′-CAAGGGCTACCATGCCAACT-3′	5′-AGGGCCAGGACCTTGCTG-3′
Human	5′-CAAGGGCTACCATGCCAACT-3′	5′-AGGGCCAGGACCTTGCTG-3′
IL-10 Mouse	5′-CTGTCATCGATTTCTCCCCTGTG-3′	5′-TGGTCTTGGAGCTTATTAAAATCAC-3′
Human	5′-CTGTCATCGATTTCTCCCCTGTG-3′	5′-TGGTCTTGGAGCTTATTAAAATCAC-3′
GAPDH Mouse	5′-AAATGGTGAAGGTCGGTGTG-3′	5′-TGAAGGGGTCGTTGATGG-3′
Human	5′-AAATGGTGAAGGTCGGTGTG-3′	5′-TGAAGGGGTCGTTGATGG-3′
iNOS Mouse	5′-GGGTCGTAATGTCCAGGAAGT-3′	5′-TCTTGGAGCGAGTTGTGGAT-3′
Human	5′-GGGTCGTAATGTCCAGGAAGT-3′	5′-TCTTGGAGCGAGTTGTGGAT-3′

## Data Availability

The data presented in this study are available in this article.

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
