# Peer review of "Stevioside Activates AMPK to Suppress Inflammation in Macrophages and Protects Mice from LPS-Induced Lethal Shock"

_molecules, 2021, doi:10.3390/molecules26040858_

Round 1
Reviewer 1 Report
The authors satisfactory addressed the critical points raised in the previous reviews of the manuscript
Author Response
Thank you for your elaborative review and your comments are valuable and helpful.
Reviewer 2 Report
The authors shoul consider the following suggestion:
Figures 1-4,8,10: the graph bar of the different treatmend should have a different fill.
Did the authors evaluate the effect of stevioside alone in MTT assay?
The in vivo used dose of 40 mg/kg is really high: how the authors selected this dose? Did they perform evaluations with other concentrations?
Reviewer 3 Report
The “SVS” abbreviation should be deleted in the text (line 34) as is not used in the text.
For sake of clarity, please resentence “These results indicated that stevioside really inhibited the expression of IL-6 transcriptionally, in agreement with results from reporter assay”(lines 94-95).
Author Response
Please see the attachment.

This manuscript is a resubmission of an earlier submission. The following is a list of the peer review reports and author responses from that submission.
Round 1
Reviewer 1 Report
The manuscript of Wei et al describes anti-inflammatory actions of food additive stevioside. Stevioside, in the concentration range of 30-120 µM inhibited LPS-induced IL-6, TNFa, iNOS and COX-2 expression while increasing IL-10 expression in different macrophage cell lines. Stevioside also protected mice against LPS-induced lethal shock.
Stevioside also inhibited activation of NFkB by LPS, and reduced total and nuclear levels of pro-inflammatory transcription factor IRF5. The anti-inflammatory effects of stevioside were attributed to its activation of AMPK, since AMPK inhibitor CC increased inflammatory response in the presence of stevioside.
The findings of the anti-inflammatory effect of stevioside in macrophages were published before (e.g. Ref 11), so the novel mechanistic aspect of the study would be the effect on IRF5 and the involvement of AMPK.
However, the evidence for AMPK involvement in the anti-inflammatory action of stevioside is very weak.
First, inhibition of LPS-induced inflammatory response is already observed at 50 µm stevioside, whereas slight increase of AMPK phosphorylation over LPS-induced level is only observed at 100µM stevioside. There was also no information whether AMPK substrate phosphorylation (e.g. acetyl-CoA carboxylase) is also increasingly phosphorylated after stevioside treatment.
The conclusion about AMPK involvement is only based on the effect of compound C. However, this substance is known to have many off-target effects, thus, any data using this drug should be confirmed using AMPK knockdown. There was also no control, that is cells treated with compound C without stevioside. Furthermore, in Fig 8d, the blots of Histone 1 are of unacceptable quality.
Overall, the data presented, in my opinion, do not firmly support the conclusion about the AMPK involvement. This, as well as general lack of novelty regarding the description of the anti-inflammatory effect of stevioside, precludes the publication of the study in Molecules.
Reviewer 2 Report
- In title, it seems that "Activated" could be changed into "Activates"/ "Protected" into "Protects".
- "RAW264.7" or "RAW 264.7", which is right?
- In the line of 159, "100ug/mL" should be corrected to "100 ug/mL".
- In the line of 231, "100ug/mL" should be corrected to "100 ug/mL".
- In the line of 403, "60h" should be corrected to "60 h".
- In the line of 416, "t-test" should be corrected to "t-test".
Reviewer 3 Report
The manuscript reports sufficient data in order to support the effect of stevioside but some points shoud be clarified:
- Figure 1: why in some figures the authors tested 3 concentration of stefioside and not for others?
- In the following figures the authors show the results concerning all the concentrations.
- MTT results should be showed.
- The in vivo used doses are high. How the authors selected them?
- -Discussion should take in account also the previous evidence concerning the antinflammatory effect of stevioside.
Reviewer 4 Report
The manuscript entitled "Stevioside activated AMPK to suppress inflammation in macrophages and protected mice from LPS-induced lethal shock" (Wei et al.) reports an extensive biological evaluation of the anti-inflammatory properties of stevioside. The authors carried out cell-based and in vivo assays that overall indicated that stevioside can suppress inflammation through different mechanisms.
The following observations have been raised during the review:
The authors should comment on the dose of stevoside used in both cell-based and in vivo tests and make clear if the concentration used would foreseen an anti-inflammatory effect in humans.
English language should be checked through the whole manuscript in order to avoid spelling and grammar errors, word repetition and cryptic sentences.
Figure 1a: delete “structure of stevioside” from the drawing
Round 2
Reviewer 1 Report
The authors tried to address the concerns raised in my original review by making changes to the text of the manuscript, and in their response letter, where they provide the explanations based mostly on the literature why their conclusion on AMPK involvement in anti-inflammatory action of stevioside may be valid.
However, none of the initial objections was experimentally addressed in order to strengthen the conclusions of the study. Therefore, in my opinion, the manuscript is not significantly improved to warrant publication in Molecules.